

# Ionospheric effects of 5–6 January 2019 eclipse over the People's Republic of China: Results from oblique sounding

Leonid F. Chernogor[1, 2, 3], Kostyantyn P. Garmash[2], Qiang Guo[3], Victor T. Rozumenko[2], Yu Zheng[1,*]

[1]College of Electronic Information, Qingdao University, 308 Ningxia Road, Qingdao, 266071, China

5   [2]Department of Space Radio Physics, V. N. Karazin Kharkiv National University, 4 Svobody Sq., Kharkiv, 61022, Ukraine

[3]College of Information and Communication Engineering, Harbin Engineering University, 145 Nantong Street, Nangang District, Harbin, 150001, China

*Correspondence to*: Yu Zheng (zhengyu@qdu.edu.cn)

**Bullet points:**

Solar eclipse was accompanied by up to ±1.5 Hz Doppler spectrum broadening and ±0.5 Hz variations in the Doppler shift, $f_D$.

Atmospheric gravity waves excited 15 min period variations in $f_D$ and 1.6 – 2.4 % perturbations in $N$.

Infrasound excited 4–5 min period variations in $f_D$ and 0.2 – 0.3 % perturbations in $N$.

20   The greatest decrease in the ionospheric electron density, $N$, attains –15 %, whereas theoretical estimate is –16 %.





**Abstract.** The paper deals with the variations in the Doppler spectra and in the relative amplitudes of the signals observed at oblique incidence over the People's Republic of China (PRC) during the partial solar eclipse of 5–6 January 2019 and on reference days. The observations were made using the multifrequency multiple path radio system for sounding the ionosphere at oblique incidence. The receiver system is located at the Harbin Engineering University campus, PRC, and 14 HF broadcasting station transmitters are used for taking measurements along the Lintong/Pucheng to Harbin, Hwaseong to Harbin, Chiba/Nagara to Harbin, Hailar/Nanmen to Harbin, Beijing to Harbin (three paths), Goyang to Harbin, Ulaanbaatar/Khonkhor to Harbin, Yakutsk to Harbin (two paths), Shijiazhuang to Harbin, Hohhot to Harbin, and Yamata to Harbin radio wave propagation paths. The specific feature of this partial solar eclipse was that it occurred during local time morning with a geomagnetic disturbance ($K_p \approx 3-$) in the background. The response of the ionosphere to the solar eclipse have been inferred from temporal variations in the Doppler spectra, the Doppler shift, and in the signal relative amplitude. The partial solar eclipse was found to be associated by Doppler spectrum broadening, up to $\pm 1.5$ Hz, alternating sign Doppler shift variations, up to $\pm 0.5$ Hz, in the main ray, and by quasi-periodic Doppler shift changes. The relative amplitude of electron density disturbances in the 15 min period atmospheric gravity wave field and in the 4–5 min period infrasound wave field is estimated to be $1.6 - 2.4$ % and $0.2 - 0.3$ %, respectively. The estimates of a maximum decrease in the electron density are in agreement with the observations.

## 1 Introduction

A solar eclipse is quite a rare natural phenomenon. The maximum phase of a total eclipse can persist for a fraction of a second to a maximum of 7 min 32 s, whereas a partial solar eclipse persists for about 2–3 h. The umbra's path width varies from ~150 km at the equator to 1,000 km at the poles, and the Moon's shadow travels at a speed of about 500 to 1,000 m/s, depending on the geographic latitude (Chernogor, 2013a).

A solar eclipse acts to significantly modify energy influx, capable of producing variations in all geophysical fields, changes in the parameters of the processes acting in the sub-systems of the Earth (internal spheres)–atmosphere–ionosphere–magnetosphere (EAIM) system, and of disrupting the existing couplings between the subsystems (Chernogor, 2003, 2011; Chernogor and Rozumenko, 2008).

The processes caused by a solar eclipse in the EAIM system are resembling the processes observed to occur at the morning-evening meridian, but they differ from the latter both quantitatively and qualitatively. First, the changes in the solar energy flux during the course of a solar eclipse occur at almost unchanging zenith angle (it changes only by $8 - 12$%). Second, the effects of a solar eclipse in the EAIM system depend significantly on season, local time, the state of atmospheric and space weather, etc. Moreover, unlike the terminator, the Moon's shadow moves at supersonic speed. All these factors make each eclipse one of a kind.



The study of the EAIM system response to a solar eclipse permits the establishment of direct and reverse, positive
and negative couplings between the subsystems, the specification of physical and chemical processes operating in the
subsystems, and the determination of a number of parameters of these processes, etc. (Chernogor, 2003, 2011; Chernogor
and Rozumenko, 2008).

Astronomers have been studied solar eclipses for thousands of years. The study of the upper atmosphere and
ionosphere began in the twentieth century, encompassing the effects that a solar eclipse has on these media.

First attempts to observe ionospheric effects arising during a solar eclipse date back to 1930 – 1940s (see, e.g.,
Chapman, 1932; Higgs, 1942). An eclipse-related distortion of the radio wave characteristics was used first, and then the
ionosonde technique.

The first collection of papers dealing with ionospheric effects of solar eclipses was published during 1950s (Beynon
and Brown, 1956).

The investigation of processes caused by solar eclipses became especially active with the advent of the space age
when a broad spectrum of rocket and satellite measurements found applications in the field. The incoherent scatter radar, the
single most powerful ground-based technique for probing geospacer, appeared at the same time.

During the 1970s and later, the study of processes caused by eclipses became more active. Astronomical, radio,
satellite, and other techniques were used for this purpose, which were described in special issues of journals (see, e.g.,
Eclipse Supplement …, 1970; Journal …, 1972) or in books (see, e.g., Anastassiades, 1970).

The studies conducted by Chandra et al. (1980, Sen Gupta et al. (1980), Deshpande et al. (1982), Rama Rao et al.
(1982), Roble et al. (1986), Salah et al. (1986), and Liu et al. (1998) during the 1980 – 1990s should be noted.

Results obtained in recent two decades are presented in the studies by Uryadov et al. (2000), Akimov et al. (2005,
2010), Burmaka et al. (2006a, 2006b), Founda et al. (2007), Afraimovich et al. (2007), Šauli et al. (2007), Jakowski et al.
(2008), Grigorenko et al. (2008), Lyashenko and Chernogor (2008), Akimov and Chernogor (2010), Chernogor (2010a,
2010b, 2012a, 2012b, 2013a, 2013b, 2016a, 2016b), Garmash et al. (2011), Marlton et al. (2016), Uryadov et al. (2016),
Verhulst et al. (2016), Stankov et al. (2017), Chernogor and Garmash (2017), Coster et al. (2017), Chernogor et al. (2019).

Recent work (Chernogor, 2012a, 2012b, 2013b, 2016a, 2016b; Marlton et al., 2016; Uryadov et al., 2016; Verhulst
et al., 2016; Chernogor and Garmash, 2017; Stankov et al., 2017; Chernogor et al., 2019; Panasenko et al., 2019) describes
ionospheric effects of the solar eclipse of 20 March 2015 in Europe.

Coster et al. (2017) analyze effects of the solar eclipse of 21 August 2017.

Guo et al. (2020) discuss the effects of the partial solar eclipse of 11 August 2018 in the ionosphere over the PRC.
The observations have been made with the coherent multi-frequency multiple path radio system, the receiver of which is
located at the Harbin Engineering University campus and 14 transmitters are situated in the PRC, Japan, Mongolia, the
Republic of Korea, and the Russian Federation. In the ionosphere, aperiodic and quasi-periodic perturbations in the electron
density, $N$, have been detected with a 1–10% amplitude and a ~10-min period, $T$, whereas the decrease in the ionospheric $E$
region electron density attains 26%, which agree well with the theoretical estimate (24%).





The solar eclipse of 21 June 2020 that occurred in the equatorial ionosphere was observed by Le et al. (2020), Zhang et al. (2020), Huang et al. (2020, 2021), Dang et al. (2020), Patel and Singh (2021), Wang et al. (2021*a*, 2021*b*),

Şentürk et al. (2021), Sun et al. (2021), Shagimuratov et al. (2021), Aa et al. (2021), Chen et al. (2021), Tripathi et al. (2022); the ionospheric effects were observed to occur with the Appleton anomaly in the background. Zhang et al. (2020) detected effects from the solar eclipse in the magnetically conjugate region.

Chernogor and Mylovanov (2022) describe the ionospheric effects from the annular solar eclipse of 10 June 2021 that occurred in the high latitudes. The atmospheric, ionospheric, and magnetic effects from this eclipse are analyzed by

Chernogor (2021*a*, 2021*b*, 2022), Chernogor and Mylovanov (2022), Chernogor and Garmash (2022).

The regular effects, such as a decrease in the electron density, and in the electron and ion temperatures, variations in the ion composition, and vertical plasma movements have so far been studied quite well (see, e.g., Akimov et al., 2005; Burmaka et al., 2006a, 2006b; Grigorenko et al., 2008; Lyashenko and Chernogor, 2008; Chernogor, 2012a, 2012b, 2013a, 2013b, 2016a, 2016b; Chernogor et al., 2019; Panasenko et al., 2019). The irregular effects, which may differ for different

eclipses, have been studied significantly less (Akimov et al., 2005; Burmaka et al., 2006a, 2006b; Grigorenko et al., 2008; Lyashenko and Chernogor, 2008; Akimov and Chernogor, 2010; Chernogor, 2010a, 2010b, 2012a, 2012b, 2013b; Garmash et al., 2011; Marlton et al., 2016; Chernogor and Garmash, 2017; Stankov et al., 2017; Coster et al., 2017). The generation of wave perturbations, which was foretold by Chimonas and Hines (1970), also belong to these effects.

At the present time, the problem of studying the response of all Earth's spheres to solar eclipses has become an

interdisciplinary subject. In addition to astronomers and physicists, meteorologists, medical doctors (ophthalmologists, optometrists, and even psychiatrists), sociologists, biologists, ecologists, etc. have joined the study of the subject.

Thus, a lot of observations have been made of the effects that solar eclipses have on the ionosphere over one hundred years of eclipse subject history. Nevertheless, the study of these effects remains an urgent problem. There are a few reasons for this. First, solar eclipses take place in the different regions of the world, whereas the physical processes operating

in the low- and high-latitude ionosphere differ considerably, and consequently the responses to solar eclipses in the low-, mid-, and high-latitude ionosphere also differ from place to place. Second, the response mentioned above is largely dependent on the state of atmospheric and space weather. Third, the ionospheric response is notably dependent on the time of the eclipse onset. Fourth, the ionospheric response depends on the magnitude of the eclipse and on its duration. Finally, the application of different techniques for probing the ionosphere permits the addition of extra information on the

ionospheric effects of solar eclipses and allows the revelation of new details in these effects. All these factors indicate the specific features of each solar eclipse. Along with regular features, the ionospheric response has specific features that are characteristic of the given solar eclipse, which explain the urgency of this work.

The purpose of this work is to present the observations of variations in the Doppler spectra and in the amplitudes of radio waves that travelled along oblique propagation paths over the PRC in the course of the partial solar eclipse of 5/6

January 2019 UT period and on the previous and next days. The description of the experiment is followed by the theoretical estimates of variations in the electron density during the solar eclipse and a comparison with the observations.





## 2 The state of space weather

A preliminary analysis of the state of space weather is needed to correctly select the effects from the solar eclipse. During the course of 3/4 January 2019, the proton density, $n_{sw}$, in the solar wind plasma exhibited a gradual increase from $2\times10^6$ m$^{-3}$ on

3 January 2019 to $30\times10^6$ m$^{-3}$ at 20:00 UT on 4 January 2019 (Figure 1), whereas during 4/5 January 2019, it showed a gradual decrease to the initial level of approximately $2\times10^6$ m$^{-3}$ (Retrieved from https://omniweb.gsfc.nasa.gov/form/dx1.html).

The flow speed, $V_{sw}$, of the solar wind plasma showed a gradual increase from 300 km/s on 4 January 2019 to 552 km/s at 17:00 UT on 6 January 2019.

The proton temperature in the solar wind plasma exhibited the greatest changes during 4/5 January 2019, when it showed an increase from $10^4$ K on 4 January 2019 to $2.9\times10^5$ K at 08:00 UT on 5 January 2019.

The increases in $n_{sw}$ and $V_{sw}$ observed to occur on 4 January 2019 resulted in an increase in the dynamic pressure, $p_{sw}$, from $0.2 - 0.3$ nPa up to $6 - 7$ nPa (see Figure 1).

The interplanetary magnetic field $B_y$ component showed variability within the ±5 nT limits during 4/5 January 2019.

The $B_z$ component exhibited variations from 0 nT to –5 nT after approximately 04:00 UT on 4 January 2019 to the end of the day, whereas the Akasofu function showed an increase to 8 GJ/s, which triggered a moderate magnetic storm that persisted from 16:00 UT on 4 January 2019 until the end of 5 January 2019, when the geomagnetic $K_p$ index attained a maximum, $K_{p\max}$, of 5, and the equatorial $D_{st}$ index attained a minimum, $D_{st\min}$, of –23 nT at approximately 16:00 UT on 5 January 2019.

The magnetic field perturbation, $K_p \approx 3$, took place during 6 January 2019, when the value of the $D_{st}$ index did exceed –10 nT.

Table 1 presents the daily 10.7 cm solar flux, which is used as a measure of solar activity.

In general, the states of solar activity and space weather were favorable for the observations of ionospheric effects from the partial solar eclipse of 6 January 2019 local time over the PRC.


Table 1. Daily $F_{10.7}$ index for the 2–8 January 2019 UT period.

| Date (January 2019) | 02 | 03 | 04 | 05 | 06 | 07 | 08 |
|---|---|---|---|---|---|---|---|
| $F_{10.7}$ | 72.7 | 70.2 | 69.1 | 68.8 | 69.6 | 69.1 | 69.0 |



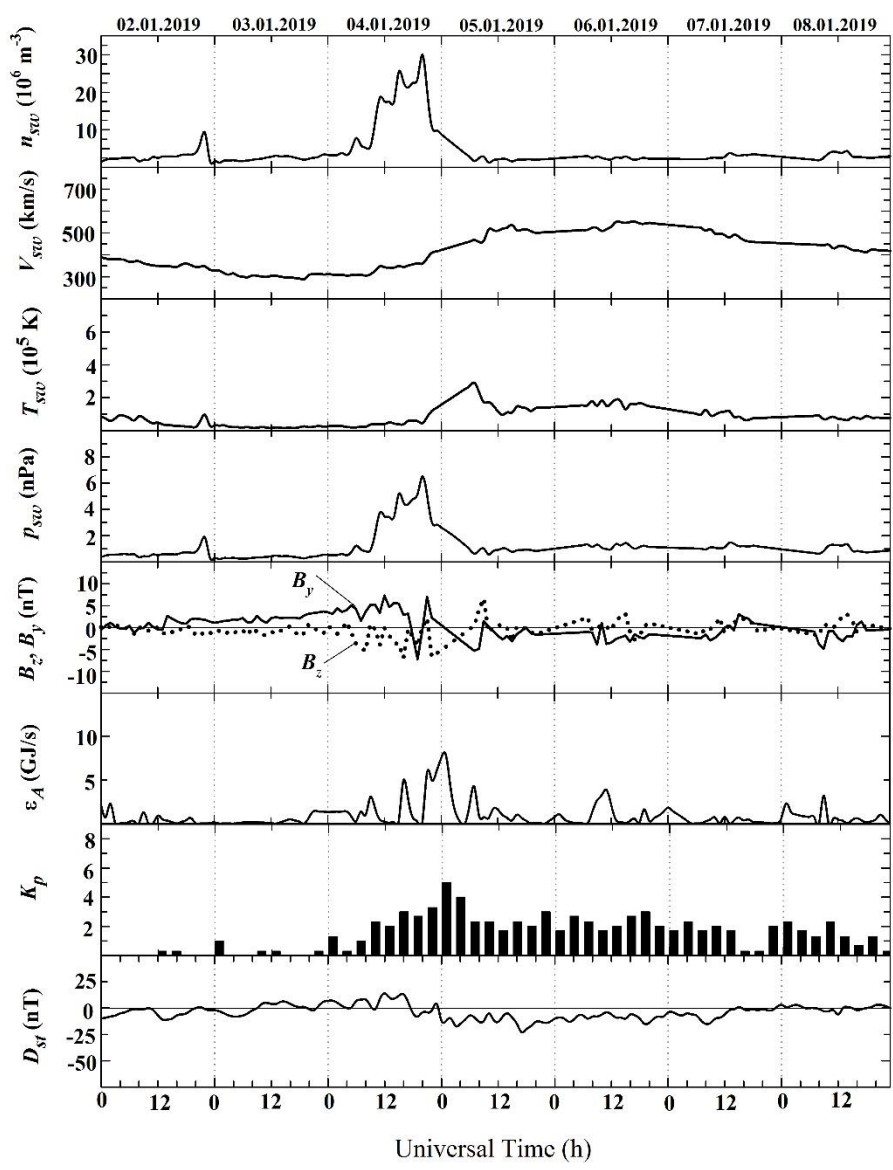

**Figure 1: Universal time dependencies of the solar wind parameters: observed proton number density $n_{sw}$, plasma speed $V_{sw}$, and temperature $T_{sw}$; calculated dynamic pressure $p_{sw}$; measured $B_z$ and $B_y$ components of the interplanetary magnetic field; estimated energy input, $\varepsilon_A$, into the Earth's magnetosphere from the solar wind per unit time; and $K_p$- and $D_{st}$-indices for the 2–8 January 2019 period. Dates are shown along the upper abscissa axis. Experimental data are retrieved from https://omniweb.gsfc.nasa.gov/form/dx1.html**




## 3 Instrumentation

The observations of the effects from the solar eclipse were made with the multi-frequency multiple path radio system designed to probe the ionosphere obliquely. The system developed in collaboration between researchers from the V. N. Karazin Kharkiv National University (Ukraine) and the Harbin Engineering University (PRC) consists of the receiver system located at the Harbin Engineering University campus (45.78°N, 126.68°E) and 14 broadcasting stations in the PRC, Japan, Mongolia, the Republic of Korea, and the Russian Federation. The system continuously monitors dynamic processes

operating in the ionosphere from May 2018 (Guo et al., 2019, 2020; Luo et al., 2020; Chernogor et al., 2020, 2021, 2022).

The receiver system is comprised of the active antenna operating in the 10 kHz – 30 MHz frequency range, the USRP N210 software defined radio using the LFRX/LRTX daughterboards, and the personal computer, for which a sophisticated software package has been developed.

This study makes use of signals that were transmitted by the broadcasting stations at Lintong/Pucheng,

Hailar/Nanmen, Beijing, Shijiazhuang, and Hohhot (PRC), Hwaseong and Goyang (Republic of Korea), Chiba/Nagara and Yamata (Japan), Ulaanbaatar/Khonkhor (Mongolia), and Yakutsk (Russian Federation), 14 propagation paths altogether, which specifications are presented in Table 2. The orientation of the propagation paths in Figure 2 shows that the propagation path midpoints were in the ionospheric regions with different Moon's coverage of the Sun.

The radio system is described in more detail by Guo et al. (2019, 2020), Luo et al. (2020), and Chernogor et al.

170 (2020, 2021).

## 4 Signal processing techniques

The information on ionospheric processes have been inferred from analysis of the temporal dependences of the Doppler spectra and the relative amplitudes of the radio waves received from all propagation paths. The spectrum content was determined over time intervals of 20.48 s by employing the autoregressive spectrum analysis of Marple (1987), that yields a

Doppler resolution of 0.01 Hz and a temporal resolution of 7.5 s. The information is derived from temporal variations in the Doppler shift, $f_D(t)$, in the main ray, from the relative amplitudes, $A(t)$, of the signals, and from Doppler spectrum broadening. Further, the Doppler shifts, $f_D(t)$, are plotted vs. time for the main ray. Next, the $f_D(t)$ time series are processed in order to determine long-term trends, short-term fluctuations, spectral content, etc.

## 5 Background information on the solar eclipse

The solar eclipse of 6 January 2019 LT was observed in Asia, viz., the PRC, Japan, the Russian Federation, the Republic of Korea, and in the North Pacific Ocean (EclipseWise.com, 2019).

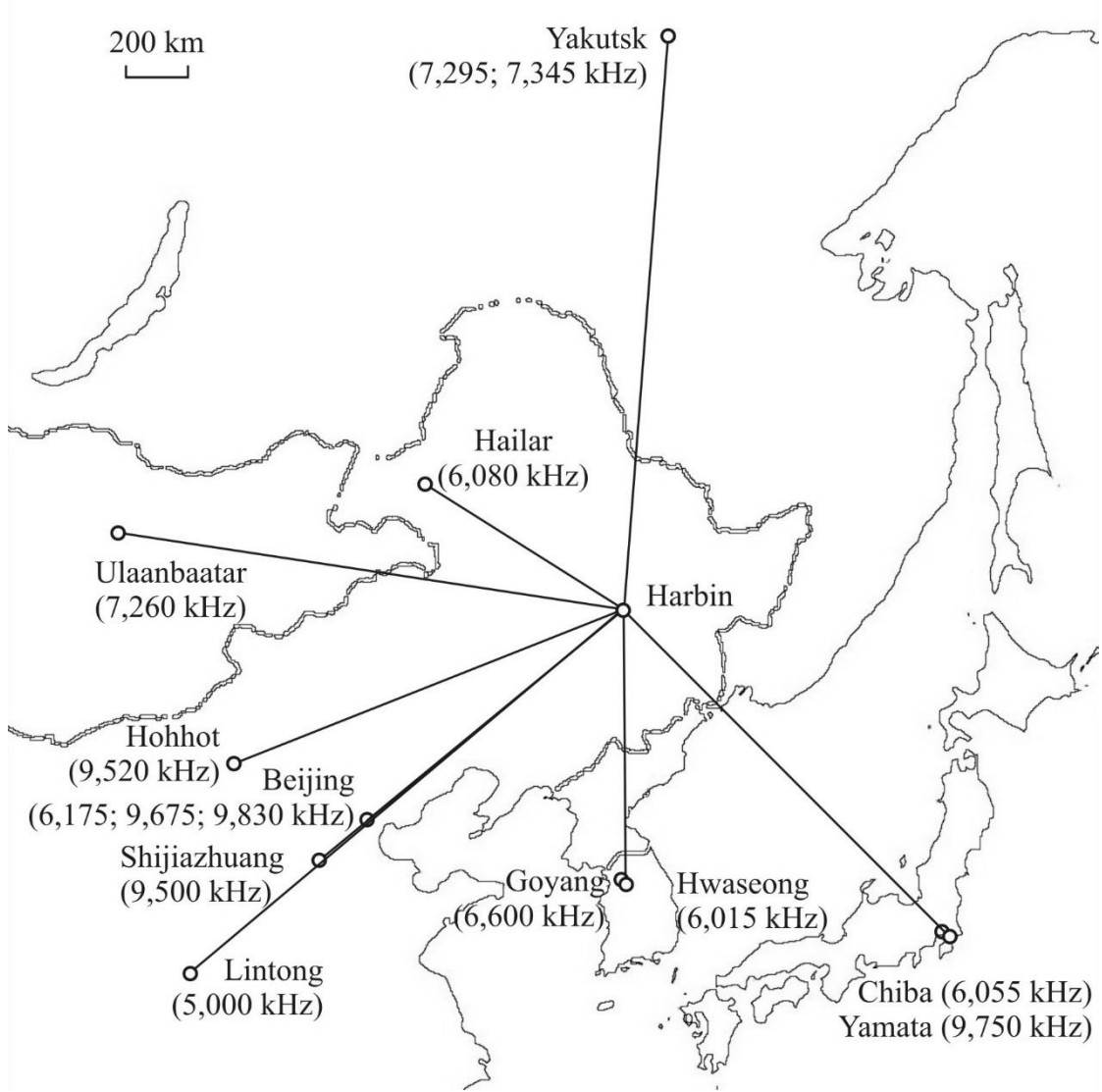

**Figure 2: Schematic diagram showing the locations of the transmitters and the receiver (Harbin) connected by the great-circle propagation paths, which were used for the observation of solar eclipse effects.**

In the PRC, the solar eclipse was observed to be partial (Figure 3). The eclipse magnitude, $M$, at an altitude of 100 km under the propagation path midpoints varied from 0.356 to 0.614, whereas the eclipse obscuration varied from 23.5 % to

51.6 % (see Table 3 where a time followed by "(r)" means the event is already in progress at sunrise). The eclipse duration changed from 133 min to 160 min. In a number of regions of the PRC, the solar eclipse began to occur prior to sunrise. The vertical lines in the middle panels in Figures 4–10 presenting the experimental data indicate the onset, the instant of greatest eclipse, and the end of the solar eclipse.



Generally, the effects from the solar eclipse in the ionosphere were observed to occur in the local time morning with

the transient processes acting in the background.

Table 2. Basic parameters of radio paths. Retrieved from https://fmscan.org/index.php.

| Transmitter | | | | Propagation path midpoints |
|---|---|---|---|---|
| Frequency (kHz) | North latitude [deg.] East longitude [deg.] | Location (country) | Distance to Harbin (km) | North latitude [deg.] East longitude [deg.] |
| 5,000 | 34.95 109.56 | Lintong/ Pucheng (China) | 1,875 | 40.37 118.12 |
| 6,015 | 37.21 126.78 | Hwaseong (Korea) | 950 | 41.50 126.73 |
| 6,055 | 35.47 140.21 | Chiba/ Nagara (Japan) | 1,610 | 40.63 133.45 |
| 6,080 | 49.18 119.72 | Hailar/ Nanmen (China) | 645 | 47.48 123.2 |
| 6,175 | 39.75 116.81 | Beijing (China) | 1,050 | 42.77 121.75 |
| 6,600 | 37.60 126.85 | Goyang (Korea) | 910 | 41.69 126.77 |
| 7,260 | 47.80 107.17 | Ulaanbaatar/ Khonkhor (Mongolia) | 1,496 | 46.79 116.93 |
| 7,295 | 62.24 129.81 | Yakutsk (Russia) | 1,845 | 54.01 128.25 |
| 7,345 | 62.24 129.81 | Yakutsk (Russia) | 1,845 | 54.01 128.25 |
| 9,500 | 38.47 114.13 | Shijiazhuang (China) | 1,310 | 42.13 120.41 |
| 9,520 | 40.72 111.55 | Hohhot (China) | 1,340 | 43.25 119.12 |
| 9,675 | 39.75 116.81 | Beijing (China) | 1,050 | 42.77 121.75 |
| 9,750 | 36.17 139.82 | Yamata (Japan) | 1,570 | 40.98 133.25 |
| 9,830 | 39.75 116.81 | Beijing (China) | 1,050 | 42.77 121.75 |

**6 Measurements and analysis**

The 4/5 and 6/7 January 2019 UT periods have been used a reference for selecting the ionospheric perturbations caused by the solar eclipse. It should be noted that the ionosphere during 4 and 5 January 2019 was partially disturbed because the

recovery phase of the moderate magnetic storm proceeded during the course of the latter days.



The measurements were made along 14 propagation paths; however, the modes of operation of only seven transmitters provided the measurements suitable for further processing.

Consider the observations and analysis in more detail.



**Figure 3: Schematic display of the Moon's shadow during the course of the partial solar eclipse of 5–6 January 2019. Retrieved from** https://eclipse.gsfc.nasa.gov/SEplot/SEplot2001/SE2019Jan06P.GIF .




Table 3. Basic information on the solar eclipse parameters at 100 km altitude over the propagation path midpoints. Retrieved from https://eclipse.gsfc.nasa.gov/JSEX/JSEX-AS.html.

| Propagation Path | Eclipse Magnitude | Eclipse Obscuration | First Contact (UT) | Sun Altitude (deg.) | Moment of Obscuration Maximum (UT) | Sun Altitude (deg.) | Sun Azimuth (deg.) | Fourth Contact (UT) | Sun Altitude (deg.) |
|---|---|---|---|---|---|---|---|---|---|
| Lintong/ Pucheng to Harbin | 0.356 | 0.235 | 23:33(r) | 0(r) | 00:35:08 | 09 | 130 | 01:46:28 | 18 |
| Hwaseong to Harbin | 0.448 | 0.326 | 23:32:46 | 04 | 00:46:40 | 15 | 139 | 02:09:54 | 23 |
| Chiba/ Nagara to Harbin | 0.483 | 0.362 | 23:35:58 | 09 | 00:56:19 | 19 | 147 | 02:26:14 | 26 |
| Hailar/ Nanmen to Harbin | 0.514 | 0.395 | 23:37(r) | 0(r) | 00:48:36 | 09 | 138 | 02:11:06 | 16 |
| Beijing to Harbin | 0.43 | 0.307 | 23:31:51 | 00 | 00:41:37 | 11 | 135 | 02:00:00 | 19 |
| Goyang to Harbin | 0.451 | 0.329 | 23:32:49 | 04 | 00:46:54 | 14 | 140 | 02:10:18 | 23 |
| Ulaanbaatar /Khonkhor to Harbin | 0.457 | 0.335 | 00:00(r) | 0(r) | 00:41:14 | 05 | 131 | 01:57:18 | 14 |
| Yakutsk to Harbin | 0.624 | 0.516 | 23:49(r) | 0(r) | 01:02:22 | 07 | 145 | 02:28:34 | 12 |
| Yakutsk to Harbin | 0.624 | 0.516 | 23:49(r) | 0(r) | 01:02:22 | 07 | 145 | 02:28:34 | 12 |
| Shijiazhuang to Harbin | 0.408 | 0.285 | 23:31:34 | 00 | 00:39:25 | 10 | 133 | 01:55:39 | 19 |
| Hohhot to Harbin | 0.416 | 0.293 | 23:38(r) | 0(r) | 00:39:14 | 08 | 132 | 01:54:58 | 17 |
| Beijing to Harbin | 0.43 | 0.307 | 23:31:51 | 00 | 00:41:37 | 11 | 135 | 02:00:00 | 19 |
| Yamata to Harbin | 0.487 | 0.367 | 23:35:56 | 09 | 00:56:15 | 19 | 147 | 02:26:07 | 26 |
| Beijing to Harbin | 0.43 | 0.307 | 23:31:51 | 00 | 00:41:37 | 11 | 135 | 02:00:00 | 19 |


## 6.1 Lintong/Pucheng to Harbin radio-wave propagation path

This radio station, operating at 5,000 kHz, is located in the PRC at a great-circle distance, $R$, of 1,875 km from Harbin. The eclipse magnitude, $M_{max}$, at an altitude of 100 km under the propagation path midpoint was estimated to be 0.356, whereas the eclipse obscuration, $B_m$, was predicted to be 0.235.

215       Figure 4 shows UT dependences of the Doppler spectra exhibiting diffuseness and occupying the frequency range from zero to 1.5 – 2.5 Hz were observed in the local time morning on both the reference days and the day when the solar eclipse occurred. On 5 January 2019, the Doppler shift in the main (maximum energy) ray first increased, fluctuating, from zero to 0.6 Hz at 00:30 UT, and then decreased to zero. The Doppler shift exhibited ~4 – 5 min period, $T$, and 0.1 Hz





amplitude, $f_{Da}$, quasi-periodic variations over the 00:50 – 01:45 UT period. The Doppler spectra showed that virtually one
ray was reflected from the ionosphere over the 01:00–03:00 UT period.

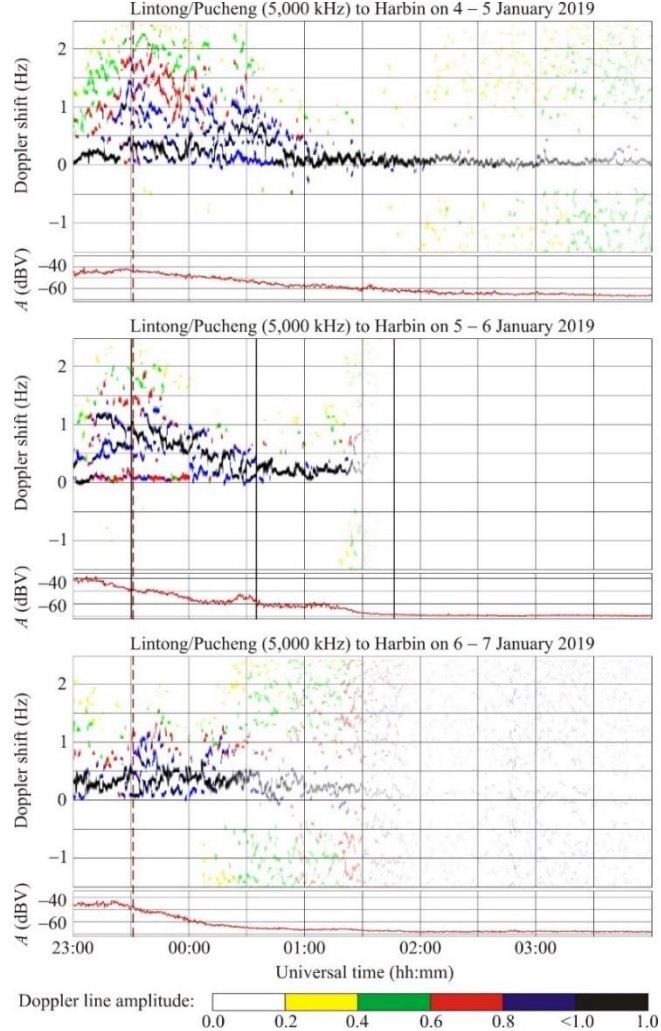

**Figure 4: Universal time variations of Doppler spectra and signal amplitude, *A*, along the Lintong/Pucheng to Harbin propagation path. The black-blue-red-green-yellow colors show the relative amplitude of 1, 0.8, 0.6, 0.4, and 0.2, respectively. Vertical solid**
**lines indicate, hereafter, the beginning, the instant of greatest eclipse, and the end of the solar eclipse at 100 km altitude. Dashed lines indicate sunrise at the ground.**

On 7 January 2019, the Doppler spectrum broadening did not exceed 1.5 Hz until 00:30 UT, and diffuseness
exhibited an increase over the 00:30 – 02:00 UT period. The Doppler shift in the main ray exhibited variations from 0.3 Hz
to 0.5 Hz, with a quasi-period, *T*, of ~25 min and amplitude, $f_{Da}$, of 0.1 Hz.





On 5 January 2019, during the 23:00 – 23:45 UT period, the Doppler spectrum broadening attained 2 Hz, whereas during the time interval from 23:45 UT on 5 January 2019 to 00:15 UT on 6 January 2019, it did not exceed 1 Hz. After this time interval, the Doppler spectra showed that one ray was reflected from the ionosphere until 01:30 UT on 6 January 2019.

Prior to the solar eclipse onset, i.e., before approximately 23:30 UT on 5 January 2019, the mean, $\overline{f_D}$, of the

Doppler shift in the main ray was observed to be approximately 1.2 Hz. Over the time interval from 23:30 UT on 5 January 2019 to 00:35 UT on 6 January 2019, the mean, $\overline{f_D}$, showed tendency to decrease from 1 Hz to 0.1 Hz, whereas $\overline{f_D} \approx 0.1 - 0.2$ Hz to the end of the eclipse. The spectrum of fluctuations in $f_D(t)$ showed oscillations with a period, $T$, of ~10 – 15 min and amplitude, $f_{Da}$, of 0.1 Hz.

In the course of the reference days, the signal amplitude, $A$, exhibited monotonous changes, viz., the amplitude

decreased by 25 dBV as the ionosphere went out of and into sunlight.

On 6 January 2019, from 00:22 UT until 00:37 UT, i.e., around the instant of greatest eclipse, the signal amplitude first showed an increase by 7 dBV and then a decrease close to the previous value.

### 6.2 Chiba/Nagara to Harbin radio-wave propagation path

This radio station, operating at 6,055 kHz, is situated in Japan at a great-circle distance, $R$, of ~1,610 km from Harbin. At the

middle of the propagation path, $M_{max} \approx 0.483$, $B_m \approx 0.362$.

During the 4/5 January 2019 UT night, from 23:00 to 01:30 UT, the Doppler spectra exhibited diffuseness (Figure 5), whereas the spectrum width attained 1.3 Hz. During the 01:30–03:40 UT period, the Doppler spectra showed that a single ray was reflected from the ionosphere, and the Doppler shift in the main ray exhibited variations from zero to 0.6 Hz.

During the 6/7 January 2019 UT reference period, the Doppler spectra showed the following behavior. From 23:00

UT to 01:30 UT, the Doppler spectrum diffuseness was observed to occur, whereas spectrum broadening attained 1 Hz, and the spectra showed that virtually a single ray was reflected from the ionosphere after 01:30 UT. The ~5 – 7 min period, $T$, and ~0.10 – 0.15 Hz amplitude, $f_{Da}$, oscillations were observed to occur in the spectrum of $f_D(t)$ in the main ray.

During the 5/6 January 2019 night, from 23:00 UT to 01:50 UT, the Doppler spectra showed diffuseness, whereas the $f_D$ values exhibited high temporal variability from –(1 – 1.5) Hz to (1 – 1.2) Hz. After the solar eclipse onset, the Doppler

spectrum width varied within the 0.5 Hz limits. From 01:00 UT, it again occupied the range from –1.5 Hz to 1.2 Hz, whereas the Doppler shift temporal dependence in the main ray showed gaps. The $\overline{f_D}$ showed a tendency to decrease from 0.5 Hz to –0.3 Hz within the 23:00 UT to 00:30 UT interval, then it was fluctuating around zero over the 00:30 – 01:00 UT period, after which the $\overline{f_D}$ exhibited a tendency to increase from 0 Hz to 0.6 Hz over a half hour interval, i.e., until 01:30 UT. Around the instant of greatest eclipse, the ~10 min period, $T$, and ~0.10 – 0.15 Hz amplitude, $f_{Da}$, oscillations were observed

to occur, while the signal amplitude exhibited an increase up to 5 dBV over about a 20 min interval. Similar effects were absent on the reference days.

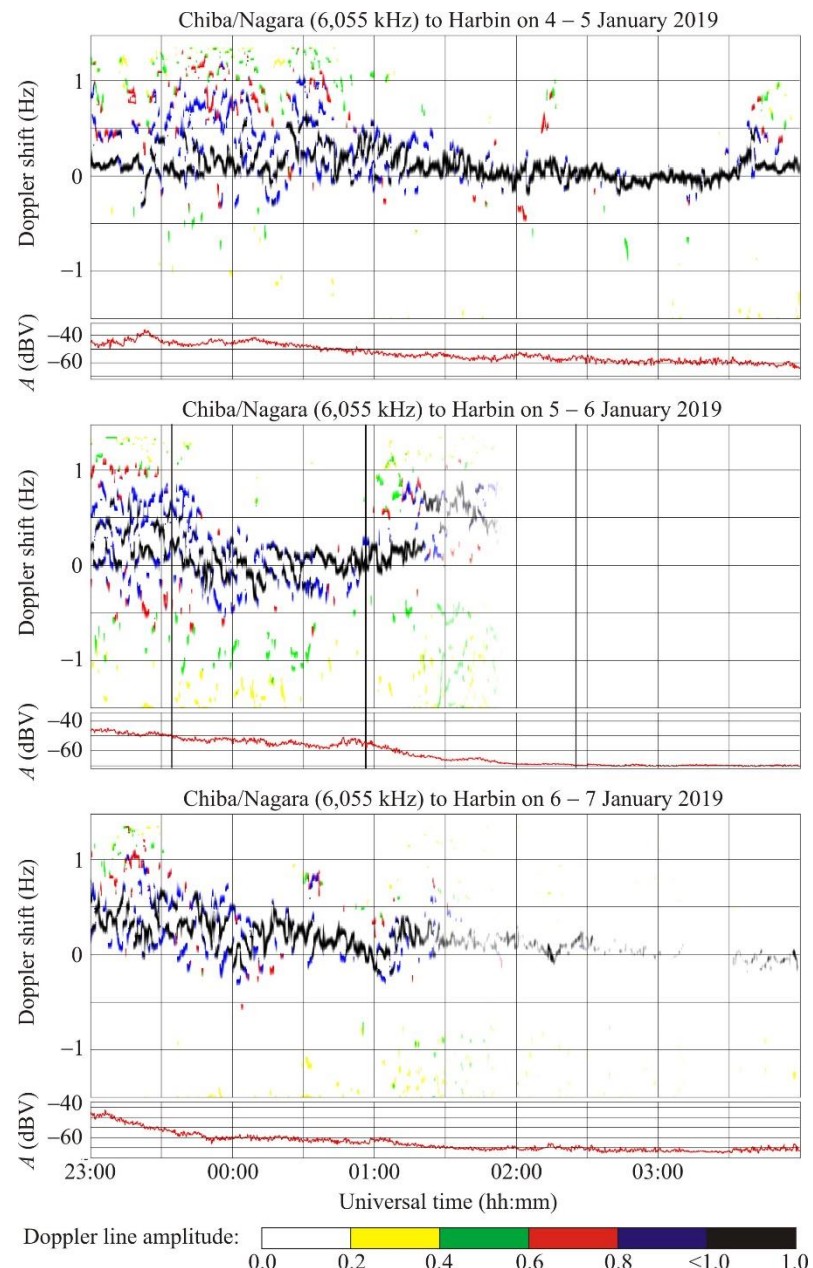

**Figure 5: The same as in Figure 4 but for the Chiba/Nagara to Harbin radio-wave propagation path.**

## 6.3 Ulaanbaatar to Harbin radio-wave propagation path

This radio station, operating at 7,260 kHz, is located in Mongolia at a great-circle range, $R$, of ~1,496 km from Harbin. The solar eclipse magnitude, $M_{max}$, at an altitude of 100 km under the midpoint of this propagation path is estimated to be ~0.457, whereas the eclipse obscuration, $B_m$, is predicted to be ~0.335.



During the 4/5 January 2019 night, from 23:00 to 01:00 UT, and from 01:50 to 02:50 UT, the Doppler spectra exhibited considerable broadening, from –1.5 Hz to 1.5 Hz (see Figure 6). The Doppler shift in the main ray showed a

decrease, time fluctuating, from 1 Hz observed at the end of 4 January 2019 to zero on 5 January 2019, whereas the range of fluctuations attained 0.20 – 0.25 Hz.

Consider the behavior of the Doppler spectra during the transition from 6 to 7 January 2019. On 6 January 2019, from about 23:20 UT to 24:00 UT, the spectra showed diffuseness and occupied the band of frequency from –(1 – 1.5) Hz to 1.5 Hz, while the main ray showed fluctuations around the –1 Hz level. On 7 January 2019, after 00:00 UT, a sharp increase

in the Doppler shift took place from –1 Hz to 0.5 Hz, and then a decrease in $f_D(t)$ was noted from 0.5 Hz to 0 Hz, while the $f_D(t)$ exhibited fluctuations within the 0.2 Hz limits.

The Doppler spectra showed broadening before the solar eclipse onset, whereas the range of fluctuations in $f_D(t)$ was observed to be close to 0.2 Hz. The transmitter was out of operation over the 00:05 – 00:50 UT period on 6 January 2019; after 00:50 UT, the Doppler spectra showed that virtually a single ray was reflected from the ionosphere, and the $f_D(t)$

showed a tendency, first, to increase to 01:15 UT, and then to decrease from 0.8 Hz to 0.2 Hz at 03:00 UT.

During the course of all days, the signal amplitude exhibited fluctuations within the 10 dBV limits. Effects of the solar eclipse were not observed reliably in the signal amplitude.

### 6.4 Shijiazhuang to Harbin radio-wave propagation path

This transmitter, operating at 9,500 kHz, is situated in the PRC at a great-circle range, $R$, of ~1,310 km from the receiver at

Harbin. The solar eclipse magnitude, $M_{max}$, at an altitude of 100 km under the midpoint of this propagation path is estimated to be ~0.408, whereas the eclipse obscuration, $B_m$, is predicted to be ~0.285.

During the 4/5 January 2019 night, from 23:20 to 00:30 UT, the Doppler spectra were observed to broaden, from 0 Hz to 1.5 Hz (see Figure 7), whereas from 23:40 to 01:00 UT, the Doppler shift in the main ray showed a decrease from 0.25 Hz to 0 Hz.

During the 6/7 January 2019 night, from 23:30 to 04:00 UT, the Doppler spectra showed that a single ray was reflected from the ionosphere, while the $f_D(t)$ exhibited a decrease, fluctuating, from 0.3 Hz to 0.1 Hz.

On the local time day, when the solar eclipse occurred, the Doppler spectra and the Doppler shifts were notably different from those observed on the reference days. On the night of 5/6 January 2019, until 23:30 UT, the fluctuations in $f_D(t)$ were insignificant since the radio waves were reflected from the sporadic $E$ layer, whereas from 23:30 UT to 01:00 UT the

radio waves were reflected from the $F$ region, and consequently the $f_D(t)$ were observed to fluctuate widely, from 0.4 Hz to – 0.2 Hz. The $\overline{f_D}(t)$ showed a tendency to decrease quasi-periodically from 0.4 Hz to 0.2 Hz over the 23:30 – 24:00 UT period, with the ~10 – 15 min period, $T$, and ~0.1 Hz amplitude, $f_{Da}$, variations being superimposed on the $\overline{f_D}(t)$.





**Figure 6: The same as in Figure 4 but for the Ulaanbaatar to Harbin radio-wave propagation path.**



**Figure 7: The same as in Figure 4 but for the Shijiazhuang to Harbin radio-wave propagation path.**






On 6 January 2019, from 00:00 to 00:30 UT, $\overline{f_D}(t) \approx 0.2$ Hz. Around the instant of the greatest occultation of the Sun's area, $\overline{f_D} \approx 0$ Hz, after which the $\overline{f_D}$ was observed to grow from nearly zero to 0.25 Hz during the course of 15 min, and then the $\overline{f_D}$ exhibited a decrease from 0.25 Hz to zero. A partial screening of reflections from the *F* layer by the sporadic *E* layer was noted after about 00:55 UT on 6 January 2019, whereas a significant (from –1.5 Hz to 1.5 Hz) broadening appeared in the Doppler spectra. Two Doppler lines, with $\overline{f_D}(t) \approx 0.1$ Hz and $\overline{f_D}(t) \approx 0.4$ Hz, were the more conspicuous in the Doppler spectrum within the 00:50 – 01:45 UT period on 6 January 2019.

The signal strengths exhibited high temporal variability (up to 12 dBV) along this propagation path during the reference days, whereas during the 5/6 January 2019 night, from 23:35 to 00:40 UT, i.e., around the instant of greatest eclipse, variations in *A*(*t*) attained 18 dBV.

**6.5 Hohhot to Harbin radio-wave propagation path**

This propagation path at ~9,520 kHz passes along the great-circle path length, *R*, of ~1,340 km in the PRC. At the transmitter location, $M_{max} \approx 0.416$, $B_m \approx 0.293$.

In the course of the 4/5 January 2019 night, the Doppler spectra showed that a single ray was reflected almost all the time from 23:40 UT to 04:00 UT (Figure 8). The spectrum broadening from 0.5 Hz to 1.5 Hz was observed to occur only over the 23:20 – 23:55 UT period. The Doppler spectra showed that two rays were reflected from the ionosphere over the 02:45 UT to 03:50 UT period, whereas the $f_D(t)$ was observed to exhibit high temporal variability from 00:00 UT to 01:00 UT.

During the course of the 6/7 January 2019 night, the Doppler spectra showed that only a single ray was reflected from the ionosphere virtually all the time from 23:00 UT to 04:00 UT, whereas the $f_D(t)$ exhibited high temporal variability over the 23:40 – 01:00 UT period.

The equality $\overline{f_D}(t) \approx 0$ Hz was observed to hold during the 5/6 January 2019 night before the solar eclipse onset, when the radio wave began reflecting from the ionospheric *F* region. During the following 30 min, the Doppler shift exhibited a decrease from 0.5 Hz to 0.1 Hz, and the $\overline{f_D}(t)$ showed a decrease from 0.1 Hz to 0 Hz during the time interval until 00:30 UT. The $f_D(t)$ exhibited high (from 0.5 Hz to 0 Hz) temporal variability over the 00:30 – 02:30 UT period, after which the range of fluctuations did not exceed 0.2 Hz. The ~15 min period, *T*, and ~0.10 – 0.15 Hz amplitude, $f_{Da}$, Doppler lines were noted during the course of the solar eclipse.

During the 23:40–00:25 UT period in the course of the solar eclipse on 5/6 January 2019 night, the signal amplitude exhibited an increase by 20 – 25 dBV, whereas the increase shown by *A* on the reference days did not exceed 10 – 15 dBV.





**Figure 8: The same as in Figure 4 but for the Hohhot to Harbin radio-wave propagation path.**



### 6.6 Beijing to Harbin radio-wave propagation path

This radio station operating at 9,675 kHz is located in the PRC at a great-circle range, $R$, of ~1,050 km from the receiver at Harbin. The solar eclipse magnitude $M_{max}$ at the location of the transmitter is estimated to be ~0.430, whereas the eclipse obscuration, $B_m$, is predicted to be ~0.307.

Figure 9 shows that the Doppler shift trend $\overline{f_D}(t) \approx 0$ until 00:30 UT on the 4/5 January 2019 night. The Doppler shift exhibits quasi-periodic variations with an ~20 min period, $T$, and an ~0.15 Hz amplitude, $f_{Da}$, over the 00:30 – 02:00 UT period. The Doppler spectra show that single rays are reflected from the ionosphere almost all the time, whereas the Doppler spectra exhibit broadening only during the 23:35 UT and 00:25 UT periods.

During the 6/7 January 2019 night, the Doppler spectra showed that single rays were reflected from the ionosphere. The $f_D(t)$ exhibited quasi-periodic variations with an amplitude of about 0.1 Hz and a period changing from 15 min to 20 min over the 23:45–01:45 UT period.

Consider the time interval around the solar eclipse on the 5/6 January 2019 night. The trend $\overline{f_D}(t) \approx 0$ until 23:50 UT. Then it showed an increase from zero to 0.4 Hz, which was followed by a decrease to zero, over the 23:50–00:20 UT period. The ~4 – 5 min period, $T$, and ~0.05 Hz amplitude quasi-periodic variations were noted in $f_D(t)$ over the 00:42–02:15 UT period. In addition, other Doppler lines 0.5 – 1 Hz apart from the main Doppler line were observed to occur over the 00:55 – 03:00 UT period.

The signal amplitude showed variability within the 3–5 dBV limits on the reference days, whereas it exhibited an increase to 10 – 20 dBV in the course of the solar eclipse.

### 6.7 Beijing to Harbin radio-wave propagation path

This radio station, broadcasting at 9,830 kHz, occupies the same site as the radio station broadcasting at 9,675 kHz; therefore, all effects observed at these two frequencies are similar, as can be seen in Figure 10.



**Figure 9: The same as in Figure 4 but for the Beijing to Harbin radio-wave propagation path.**

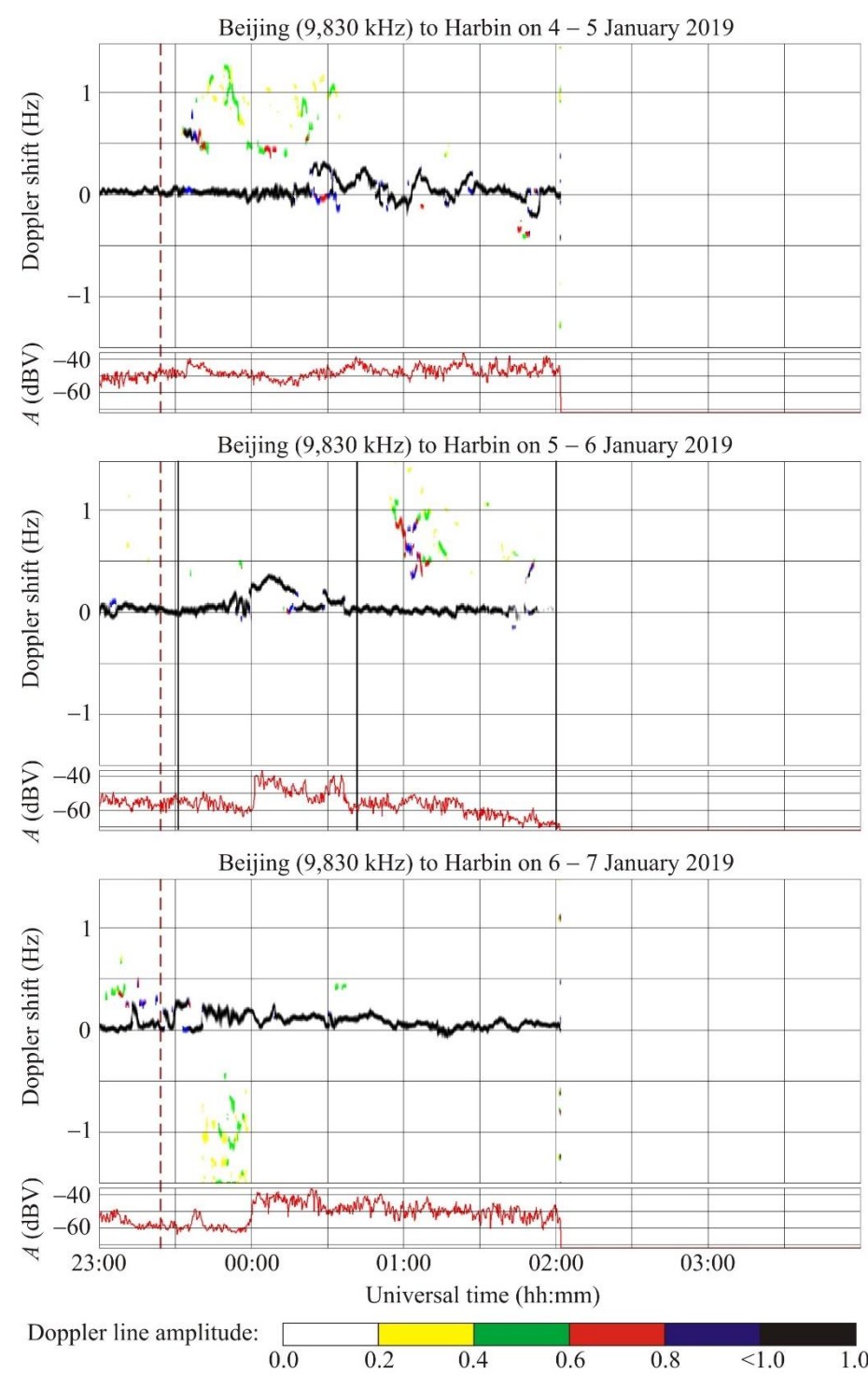

**Figure 10: The same as in Figure 4 but for the Beijing to Harbin radio-wave propagation path.**






## 7 Discussion

### 7.1 Effects from solar eclipses

Solar eclipses are well known for having the capability to cause significant variations in the parameters of both the medium and the geophysical fields in all subsystems in the Earth–atmosphere–ionosphere–magnetosphere system (see, e.g., Chernogor, 2003, 2011, 2013a; Chernogor and Rozumenko, 2008). Solar eclipses act to notably cool the air and the ground surface, to decrease the air pressure, etc. The plasma parameters and dynamics in the ionosphere show significant changes: the electron density decreases, the ion and electron temperatures reduce, the rates of chemical reactions alter, the settled

coupling between the ionosphere and the plasmasphere becomes disturbed, the precipitation of energetic electrons is made possible from the magnetosphere into the atmosphere causing additional ionization (Chernogor, 2013a). Özcan and Aydoğdu (2004) and Chernogor (2012, 2013a) describe the effects of solar eclipses also in the geomagnetic field.

The regular effects of solar eclipses may be regarded as quite well known, whereas irregular effects and their influence on the propagation of radio waves in various frequency bands have been studied to a lesser degree. On the one

hand, the irregular effects in the Earth–atmosphere–ionosphere–magnetosphere system are associated with the generation of different kinds of instability in the atmosphere and in its plasma component, turbulence production in these media, the amplification and generation of the waves of various nature (infrasound, atmospheric gravity waves, magnetohydrodynamic waves, etc.) On the other hand, the irregular effects significantly depend on the subsystem perturbation in the Earth–atmosphere–ionosphere–magnetosphere system, on the state of atmospheric and space weather, season, local time, the

magnitude of the solar eclipse, and on the geographic coordinates of the observation site. The scientific studies conducted for many years have shown that the responses of the subsystems in the Earth–atmosphere–ionosphere–magnetosphere system to the solar eclipses that occurred over the period from the end of the twentieth century to the beginning of the twenty-first century have never been the same (Chernogor, 2013a).

### 7.2 Basics of the variations in the Doppler shift during the course of solar eclipses

It is known that for a plane stratified isotropic ionosphere the Doppler shift is given by (see, e.g., Davies, 1990)

$$f_D = -\frac{f}{c}\frac{d}{dt}\int_s n\, ds = -2\frac{f}{c}\int_0^{z_r} n\sec\vartheta\, dz \,, \tag{1}$$

where $f$ is the frequency, $c$ is the speed of light, $t$ is time, $n$ is the refractive index, $ds = \sec\vartheta\, dz$ is a path element, $\vartheta(z)$ is the angle that the ray makes with the vertical, $z_r$ is the altitude of reflection.

Taking into account the dependence

$$n = n\big(N\big(\chi(t), B(t)\big)\big),$$

in (1) yields





$$f_D = -\frac{f}{c}\int_s \frac{dn}{dN}\frac{dN}{dt}\,ds = -\frac{f}{c}\int_s \frac{dn}{dN}\left(\frac{\partial N}{\partial \chi}\frac{d\chi}{dt} + \frac{\partial N}{\partial B}\frac{dB}{dt}\right)ds\ . \tag{2}$$

Here, $N$ is the electron density, $\chi(t)$ is the solar zenith angle, $B = S(t)/S_0$ is eclipse obscuration, $S_0$ is the Sun's surface area, $S(t)$ is the Sun's surface area occulted by the Moon. Equation (2) suggests that the behavior of $f_D(t)$ depends on the rate of change

of $\partial N/\partial \chi$, $d\chi/dt$, $\partial N/\partial B$, and $dB/dt$ with time. It should be noted that $dn/dN < 0$ for a plasma. In addition, $\partial N/\partial \chi < 0$, $\partial N/\partial B < 0$. During the morning hours, $d\chi/dt < 0$, whereas $d\chi/dt > 0$ during the course the afternoon hours.

First, consider the classical picture of the behavior of the Doppler shift $f_D(t)$, which take place around the local noon, $d\chi/dt \approx 0$, in the absence of fluctuations in the parameters of the medium. In this case, the magnitude and sign of $f_D(t)$ is dependent only on the multiplier $dB/dt$.

The equality $dB/dt = 0$ holds before the solar eclipse onset ($t = t_1$), at the moment, $t_m$, when the coverage of the Sun's surface area by the Moon is a maximum, and after the solar eclipse ceases to exist ($t = t_2$). If $t < t_m$, $dB/dt > 0$, whereas if $t > t_m$, $dB/dt < 0$. Note that the $f_D(t)$ is a two-lobe dependence symmetrical about the moment, $t_m$, of maximum coverage of the Sun's surface area by the Moon. Within the first lobe $t < t_m$, $f_D(t) < 0$; and within the second one $t > t_m$, $f_D(t) > 0$. At $t = t_m$, the value of $f_D(t_m)$ vanishes. It is important that $|f_{D\min}| = f_{D\max}$.

Away from the local noon, the magnitude and sign of $f_D(t)$ is determined by the relation between the expressions $(\partial N/\partial \chi)\cdot(d\chi/dt)$ and $(\partial N/\partial B)\cdot(dB/dt)$ and by the signs of $d\chi/dt$, $dB/dt$. At the stage when the occultation of the solar disk is increasing, $f_D < 0$, whereas at $t = t_m$, $f_D = 0$. At the stage when the occultation of the solar disk is decreasing, $f_D > 0$. The Doppler shift, $f_D(t)$, shows a minimum, $f_{D\min}$, at the moment $t = t_{\min}$, which is found within the interval $t_1 < t_{\min} < t_m$; whereas the Doppler shift, $f_D(t)$, shows a maximum, $f_{D\max}$, at the moment $t = t_{\max}$, which is found within the interval $t_m < t_{\max} < t_2$. It is

important that $|f_{D\min}| \neq f_{D\max}$.

If a solar eclipse begins during the morning hours, the first term in the integrant of (2) is positive, and its value can become equal to or greater than the value of the second term, which is negative if $t_1 < t < t_m$. As a result, the effect of suppression arises in the response of the Doppler shift to the solar eclipse when an increase in $N$ in the morning partially or totally compensate a decrease in $N$ due to the screening out the solar disk. During the $t_m < t < t_2$ interval, $dB/dt < 0$, and both

the terms in the integrant of (2) are positive, and consequently an increase in the Doppler shift occurs.

If a solar eclipse begins, $t_1 < t < t_m$, during the local afternoon when $d\chi/dt > 0$, and both the terms in the integrant of (2) are negative, the Doppler shift exhibits the effect of amplification. If a solar eclipse ceases to exist, $t_m < t < t_2$, during the local afternoon, the terms in the integrant of (2) have different signs (the first one is negative, and the second one is positive), the Doppler shift shows the effect of suppression.

Thus, the real $f_D(t)$ dependence may significantly differ from the classical one. In addition, the fluctuations and quasi-periodic variations in the Doppler shift are superimposed on the aperiodic changes in the trend $\overline{f_D}(t)$.





### 7.3 Influence of the solar terminator on the Doppler shift

This solar eclipse took place before or immediately after the sunrise, which is one of its features; therefore, the effects of the dawn terminator could be significant. The Doppler spectra show that the movement of the solar terminator was accompanied
by significant diffuseness along virtually all propagation paths, whereas the Doppler spectra broadening attained 1.5 – 2.5 Hz. In addition, the Doppler shift exhibited noticeable variations in the main ray. However, the effects mentioned above were absent along both the Beijing to Harbin propagation paths because, most probably, sporadic $E$ propagation took place along these propagation paths, where $f_D(t) \approx 0$ Hz and diffuseness in the Doppler spectra was absent.

The analysis of UT dependences of Doppler spectra has shown that the effects of the terminator ceased to exist
either before the beginning of the solar eclipse or soon after this moment, which made the detection of the solar eclipse effects easier to perform. At the same time, an increase in $N(t)$ that followed the terminator made the response to the solar eclipse more difficult to find.

### 7.4 Connection of the variations observed in the Doppler shifts with the solar eclipse

### 7.4.1 Lintong/Pucheng to Harbin radio-wave propagation path

Figure 4 shows that the Doppler shift in the main ray exhibits a tendency to decrease from 0.8 Hz to 0.2 Hz after the solar eclipse onset. After $t > t_m$, the $f_D$ is noted for an increase from 0.2 Hz to 0.4 Hz, and then for its reduction to the initial value. Since the analogous variations are absent on the reference days, they may be suggested to be due to the solar eclipse; most probably, an increase of 6 dBV in the signal amplitude is associated with the solar eclipse as well.

### 7.4.2 Chiba/Nagara to Harbin radio-wave propagation path

The Doppler shift of frequency showed a tendency to decrease from 0.2 – 0.4 Hz to –(0.2 – 0.4) Hz immediately after the solar eclipse onset, which was followed by an increase to zero, during the course of about 1 h (see Figure 5). After the moment, $t_m$, when the coverage of the Sun's surface area by the Moon was a maximum, i.e., when $t > t_m$, the $f_D$ showed a tendency for an increase during 30 min. In addition, the Doppler spectra exhibited considerable, from –1.5 Hz to 1.2 Hz, broadening over the 6 January 2019 01:00 – 01:50 UT period, whereas similar effects were absent on the reference days. The
effects described above may be supposed to be due to the solar eclipse.

### 7.4.3 Ulaanbaatar/Khonkhor to Harbin radio-wave propagation path

The rise followed by the fall in the Doppler shift observed during the 00:50 – 03:00 UT period on 6 January 2019 could be caused by the solar eclipse (see Figure 6), since analogous behavior of $f_D(t)$ was not observed within this time interval on the reference days. The diffuseness shown by the Doppler spectra from 00:50 UT to 01:25 UT on 6 January 2019 period is
probably also associated with the solar eclipse.





### 7.4.4 Shijiazhuang to Harbin radio-wave propagation path

Figure 7 shows that $f_D \approx 0$ Hz prior to the solar eclipse onset, whereas at about 23:30 UT on 5 January 2019, the $f_D$ exhibits an abrupt increase of 0.4 Hz, and then a tendency for 0.2 Hz reduction, which persists for about 30 min. During the following 40 min, the trend $\overline{f_D} \approx 0.2$ Hz. Around $t \approx t_m$, $\overline{f_D} \approx 0$ Hz, whereas after 00:45 UT (on 6 January 2019), the $f_D$

exhibits an increase from 0 Hz to 0.25 Hz, and then a reduction to zero, which continue for about 12 min. The Doppler spectra show diffuseness, and the Doppler shift exhibits temporal variability from –1.5 Hz to 1.5 Hz over the 00:50 to 04:00 UT period on 6 January 2019. During the 6 January 2019 00:50 – 02:13 UT period, the second powerful enough ray was sporadically appearing with the ~0.3 Hz trend, $\overline{f_D}(t)$, whereas $\overline{f_D} \approx 0$ Hz for the first ray. Some radio wave energy was most likely leaked through the screening sporadic $E$ layer.

460   The signal amplitude is observed to increase by 12 – 18 dBV during the 5/6 January 2019 night, from 23:35 to 00:50 UT, whereas on the reference days, the variations in $A(t)$ do not exceed 12 dBV.

  There are reasons to consider the effects described above to be due to the solar eclipse.

### 7.4.5 Hohhot to Harbin radio-wave propagation path

Prior to the solar eclipse onset, the Doppler shift, $f_D$, was nearly zero, since the radio waves were apparently reflected from

the sporadic $E$ layer (see Figure 8). At 03:40 UT on 5 January 2019, the radio wave penetrated into the ionospheric $F$ region, and the trend $\overline{f_D} \approx 0.5$ Hz. This moment virtually coincided with the sunrise and with the onset of the observable solar eclipse. Afterwards, a gradual decrease in $\overline{f_D}$ to zero was noted to persist during a 40 min interval. Around 00:20 UT on 6 January 2019, $\overline{f_D} \approx 0$ Hz, which was followed by an increase in $\overline{f_D}$ to 0.4 Hz and by a decrease to zero at 01:30 UT on 6 January 2019. The quasi-periodic ~15 min period, $T$, and ~0.10 – 0.15 Hz amplitude, $f_{Da}$, variations were superimposed on

the regular temporal variation in $\overline{f_D}(t)$. Other, weaker, rays were noted during the course of the solar eclipse and during 1 h interval after the solar eclipse.

  Thus, the Doppler the spectra on the day when the solar eclipse occurred were significantly different from those observed on the reference days. The behaviors of the signal amplitudes were also significantly different. All these features support the idea that the effects described above are most likely associated with the solar eclipse. At the same time, the $f_D(t)$

dependences along all propagation paths exhibit behaviors that are significantly different from the classical behavior.

### 7.4.6 Beijing to Harbin radio-wave propagation path

Figures 9 and 10 show that UT dependencies of the Doppler spectra and signal amplitudes at 9,675 kHz and 9,830 kHz were observed to be virtually the same. Therefore, the use of only the first frequency is enough for the description of the effects.





Prior to the solar eclipse onset and during 1 h interval after it, the trend $\overline{f_D} \approx 0$ Hz; the radio waves were most
likely reflected from the sporadic $E$ layer. The reflection from the ionospheric $F$ region took place only during the 5/6
January 2019 night, from 23:50 to 00:20 UT when $\overline{f_{D\,max}} \approx 0.4$ Hz. For this reason, the effect of the solar eclipse was
masked. The appearance of rays showing the ~0.2 – 1 Hz and ~0.2 – 0.7 Hz trends, $\overline{f_D}$, over the 00:55 – 01:15 UT and
01:42 – 02:30 UT periods, respectively, are most likely associated with the solar eclipse. In addition, the $f_D(t)$ exhibited weak
quasi-periodic ~4 – 5 min period, $T$, and ~0.05 Hz amplitude, $f_{Da}$, variations during all the course of the solar eclipse,
whereas the ~20 min period, $T$, and ~0.2 Hz amplitude, $f_{Da}$, oscillations were predominant on the reference days.

The signal amplitude exhibited an increase of 10 – 20 dBV during the course of the solar eclipse, whereas on the
reference days the increase did not exceed 5 – 10 dBV.

## 7.5 Results of calculations

### 7.5.1 Decrease in the electron density during the solar eclipse

Making use of the continuity equation for the electron density in the altitude range where the molecular ions are dominant,
and taking into account the production rate that is due only to photoionization by photons from the solar disk, yields

$$\frac{N(t)}{N_0} = \sqrt{1 - B(t)} \,,$$

where $N_0$ is the electron density in the absence of the solar eclipse. The value $B(t_m) = B_m$ yields the maximum effect

$$\frac{N_{min}}{N_0} = \sqrt{1 - B_m} \,. \tag{3}$$

Substituting the values $B_m \approx 0.235 - 0.362$ observed at the propagation path midpoints into Equation (3) yields
$N_{min}/N_0 \approx 0.88 - 0.80$, $\Delta N_{max}/N_0 \approx 12 - 20\%$, where $\Delta N_{max} = N_0 - N_{min}$.

### 7.5.2 Estimates of a decrease in the electron density from observational data

Guo et al. (2019, 2020), Luo et al. (2020), and Chernogor et al. (2020) have replaced the actual trajectory by two
straightened line segments intersecting at the height of reflection $z_r$, ignored the geomagnetic field, and have obtained the
following relation for estimating $\Delta N_{max}/N_0$, if the trend changes from $\overline{f_D}$ to $\Delta\overline{f}_{Dmax}$ during the time interval $\Delta t$:

$$\frac{\Delta N_{max}}{N_0} = \frac{c\Delta t}{4L_n} \frac{\kappa^2}{\kappa_\theta} \frac{\Delta\overline{f}_{D\,max}}{f} \,, \tag{4}$$

where $L_n = z_r - z_0$, $z_r$ is the reflection height, $z_0$ is the altitude of the beginning of the layer contributing to the Doppler shift,

$$\kappa^2 = \frac{1}{1 + 2\zeta\tan^2\theta} \,, \quad \zeta = \frac{z_r - z_0}{r_0} \,, \quad \tan\theta = \frac{R}{2z_r} \,, \tag{5}$$





$$\kappa_\theta = \frac{\cos\theta}{1+\sin\theta}. \tag{6}$$

Here $r_0 \approx 6,400$ km is the Earth's radius, $\theta$ is the angle of incidence with respect to the vertical at the basis of the ionosphere.

Consider, for example, the Hohhot to Harbin 9,520 kHz propagation path. The Doppler shift exhibited a maximum, $\Delta f_{D\max}$, of ~0.4 Hz during the 23:40 UT to 00:10 UT period and a minimum, $\Delta f_{D\min}$, of –0.4 Hz during the 00:30 – 01:00 UT interval on the 5/6 January 2019 night. Assuming $z_r \approx 220$ km and $z_0 \approx 160$ km yields $\theta \approx 70.3°$, $L_n \approx 140$ km, whereas substituting these numbers in (5) and (6) gives $\kappa^2 \approx 0.63$, $\kappa_\theta = 0.175$, and finally putting the latter values in (4) now yields

$$\left(\frac{\Delta N}{N_0}\right)_{\max} \approx -0.15. $$

This experimental estimate can be compared with the theoretical estimate. Substituting the eclipse obscuration, $B_m$, of ~0.293 estimated to be at the Hohhot to Harbin propagation path midpoint into (3) yields $N_{\min}/N_0 \approx 0.84$, and $(\Delta N/N_0)_{\max} \approx -0.16$, and hence this theoretical value shows a good agreement with the experimental estimate of –0.15 obtained as described above.

**7.5.3 Estimates of wave perturbation amplitudes in the atmospheric gravity wave range**

Most of the solar eclipses are well known for their capability to generate or amplify atmospheric gravity waves in the 10 – 120 min period range (see, e.g., Burmaka et al., 2006a, 2006b; Šauli et al., 2007; Chernogor, 2010, 2012, 2016a, 2016b). These waves act to excite traveling ionospheric disturbances of the same periods. Within the data segment under the study in this piece of research, oscillations in $f_D(t)$ with a period, $T$, of ~15 – 20 min are also observed along a number of the

propagation paths.

The estimate of the relative disturbance, $\delta_{Na}$, in the electron density at the reflection height, $z_r$, can be obtained from the expression analogous to (4) (Guo et al., 2019, 2020; Luo et al., 2020, Chernogor et al., 2020):

$$\delta_{Na} = \frac{cT}{4\pi L}\frac{\kappa^2}{\kappa_\theta}\frac{f_{Da}}{f}, \tag{7}$$

where

$$L = \frac{2HL_n}{L_n + 2H}, \tag{8}$$

$H$ is a scale height of the atmosphere. Assuming $2H \approx 80$ km, $L_n \approx 140$ km around the reflection height $z_r$, in (8) yields $L \approx 50$ km. Substituting $T \approx 15$ min and $f_{Da} \approx 0.10 – 0.15$ Hz, observed along the Hohhot to Harbin propagation path, into (7) yields $\delta_N(z_r) \approx 1.6 – 2.4\%$.



### 7.5.4 Estimates of wave disturbance amplitudes in the infrasound period range

Infrasound waves of great enough periods (1 – 5 min) reach the ionospheric $F$ region altitudes, modulate the electron density $N$, and consequently $f_D(t)$ (see, e.g., Gossard and Hooke, 1975; Guo et al., 2019, 2020; Chernogor et al., 2020).

Consider, for example, the Beijing to Harbin radio-wave propagation path. Assuming $z_r \approx 220$ km, $z_0 \approx 160$ km yields $\zeta \approx 10^{-2}$, $\theta \approx 67.3°$, $\kappa^2/\kappa_\theta \approx 3.59$. Substituting $L \approx 50$ km and $f_{Da} \approx 0.05$ Hz into (7) gives $\delta_N \approx 0.2 - 0.3\%$ for $T = 4 - 5$ min.

### 7.5.5 Comparisons with the effects from the solar eclipse of 11 August 2018 that took place in the People's Republic of China


The solar eclipses of 11 August 2018 and of 5/6 January 2019 took place in the People's Republic of China. Both eclipses had close magnitudes, $M_{max}$, and obscurations, $B_m$. The effects from both eclipses were revealed with the Harbin Engineering University multi-frequency multiple path radio system.

The difference is as follows. The solar eclipse of 11 August 2018 occurred in the evening hours, whereas the solar eclipse of 5/6 January 2019 was observed in the morning hours. The effects from both eclipses were partially suppressed by the processes acting at sunset or sunrise.

In both cases, the solar eclipse was accompanied by Doppler spectrum broadening, alternating sign variations in Doppler shifts in the main rays, and by the generation of infrasound and atmospheric gravity waves.

The amplitudes of the generated waves were comparable, whereas the reductions in the electron density on a relative scale near the moment of maximum occultation of the solar disk were observed to be –26% and –15%, respectively.

### 8 Conclusions

(1) Temporal variations in the Doppler spectra and Doppler shift in the main rays, as well as in the signal amplitudes observed along seven radio-wave propagation paths, with various orientations of these paths, have been studied with the

Harbin engineering university multiple path multi-frequency radio system on the day of when the solar eclipse occurred and on the reference days. The transmitters located in Japan, Mongolia, and the PRC are sounding the ionosphere at 5,000 kHz to 9,830 kHz frequencies.

(2) The solar eclipse was accompanied by Doppler spectrum broadening, up to $\pm1.5$ Hz, by alternating sign Doppler shift variations, up to $\pm0.5$ Hz, in the main ray, and by quasi-periodic Doppler shift changes.

(3) Using alternating sign Doppler shift variations during the period of the maximum occultation of the Sun's surface area, the greatest decrease in the electron density has been estimated to be about –15 %, whereas the theoretical model has shown that it is –16 %, which may be considered as being in good agreement.

(4) The atmospheric gravity waves launched by the solar eclipse acted to excite quasi-periodic, 15 min period variations in the Doppler shift, while the amplitude of the perturbations in the electron density has been estimated to be 1.6 – 2.4 %.



(5) The infrasound waves launched by the solar eclipse acted to excite quasi-periodic, 4–5 min period variations in the Doppler shift, whereas the amplitude of the perturbations in the electron density has been estimated to be about 0.2 – 0.3 %.

**Code availability.** Software for Passive 14-Channel Doppler radar may be obtained from the website at https://dataverse.harvard.edu/dataset.xhtml?persistentId=doi:10.7910/DVN/MTGAVH (Garmash, 2021).

**Data availability.** The data sets discussed in this paper may be obtained from the website at
https://dataverse.harvard.edu/dataset.xhtml?persistentId=doi:10.7910/DVN/RMFBRN (Garmash, 2022)

**Author contribution.**

"**Conceptualization:** Leonid Chernogor; **Data Curation**: Qiang Guo and Kostiantyn Garmash; **Formal Analysis**: All Authors: Yu Zheng, Leonid Chernogor, Kostiantyn Garmash, Qiang Guo, Victor Rozumenko; **Funding Acquisition**: Qiang Guo; **Investigation**: Qiang Guo and Kostiantyn Garmash; **Methodology**: Leonid Chernogor; **Project Administration**:
Qiang Guo; **Resources**: Qiang Guo; **Software**: Kostiantyn Garmash; **Supervision:** Leonid Chernogor; **Validation**: All Authors: Leonid Chernogor, Kostiantyn Garmash, Qiang Guo, Victor Rozumenko, Yu Zheng; **Visualization**: Kostiantyn Garmash, Yu Zheng; **Writing – original draft**: All Authors: Leonid Chernogor, Kostiantyn Garmash, Qiang Guo, Victor Rozumenko, Yu Zheng; **Writing review & editing**: All Authors: Leonid Chernogor, Kostiantyn Garmash, Qiang Guo, Victor Rozumenko, Yu Zheng.

**Competing interests. The authors declare that they have no conflict of interest**

**Acknowledgments.**

This article makes use of data on the solar eclipse of 5–6 January 2019 retrieved from NASA Goddard Space Flight Center Eclipse Web sites https://eclipse.gsfc.nasa.gov/SEsaros/SEsaros.html, https://eclipse.gsfc.nasa.gov/SEplot/SEplot2001/SE2019Jan06P.GIF, and https://eclipse.gsfc.nasa.gov/JSEX/JSEX-AS.html. The
solar wind parameters were retrieved from the Goddard Space Flight Center Space Physics Data Facility https://omniweb.gsfc.nasa.gov/form/dx1.html .Work by Qiang Guo and Yu Zheng was supported by the National Key R&D Plan Strategic International Science and Technology Cooperation and Innovation (2018YFE0206500). Work by L. F. Chernogor was supported by the National Research Foundation of Ukraine for financial support (project 2020.02/0015, "Theoretical and experimental studies of global disturbances from natural and technogenic sources in the Earth-atmosphere-
ionosphere system"). Work by L. F. Chernogor was supported by the Ukraine state research projects #0121U109881 and #0121U109882. Work by K. P. Garmash was supported by the Ukraine state research project #0121U109882. Work by V. T. Rozumenko was supported by the Ukraine state research project #0121U109881.





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
