# Peer review of "Ionospheric effects of 5–6 January 2019 eclipse over the People's Republic of China: Results from oblique sounding"

_Annales Geophysicae, 2022_

## Referee Comment (RC2)

[referee-annotated manuscript omitted]

---

## Author Comment (AC1)

**RC1**: 'Comment on angeo-2022-15', Anonymous Referee #1, 10 Jun 2022  reply
23 May 2022

**Ionospheric effects of 5–6 January 2019 eclipse over the People's Republic of China: Results from oblique sounding**

Leonid Chernogor, Kostyantyn Garmash, Qiang Guo, Victor Rozumenko, and Yu Zheng

**Preprint under review for ANGEO** (discussion: open, 1 comment)
Short summary

Reply to Anonymous Referee #1

Dear Anonymous Referee #1,

Thank you very much for your valuable comments that have helped the Authors greatly improve the draft of their paper.

Your comments are placed together with the Authors answers, whereas the changes made in the text of the draft of the paper are marked in yellow.

Authors.

**Short summary**
The solar eclipse of 5–6 January 2019 perturbed the ionospheric electron density, N, observed with the receiver at the Harbin engineering university and 14 HF broadcasting stations ~1,000 km around. It was accompanied by ±1.5 Hz Doppler spectrum broadening, ±0.5 Hz Doppler shift, fD, variations, 15 min period variations in fD caused by 1.6–2.4 % perturbations in N, and 4–5 min period changes in fD caused by 0.2–0.3 % disturbances in N. The decrease in N attained ~15 % (vs modeled 16 %).

**Dear Anonymous Referee #1**. Regarding your short summary, you have helped us to have specified the decrease in the *N* marked in yellow below:

The solar eclipse of 5–6 January 2019 perturbed the ionospheric electron density, N, observed with the receiver at the Harbin engineering university and 14 HF broadcasting stations ~1,000 km around. It was accompanied by ±1.5 Hz Doppler spectrum broadening, ±0.5 Hz Doppler shift, fD, variations, 15 min period variations in fD caused by 1.6–2.4 % perturbations in N, and 4–5 min period changes in fD caused by 0.2–0.3 % disturbances in N. The decrease in N attained ~15 % (vs modeled 13 %).

- General comments:

  This paper presents Doppler spectra from oblique ionospheric soundings over various distances during a solar eclipse, as well as some theoretical calculation to derive the electron density decrease from the data. This is interesting work and, to the best of my knowledge, not published before. Thus, it should certainly be acceptable for publication in this journal. There are only a few points which I thin the authors should address a little more carefully:

  Specific comments:

  1) There seems to be some confusion throughout the paper on what altitudes are involved. The general background description seems to be considering the eclipse at sea level (e.g. Figure 3 is valid for sea-level). But, In the caption to Figure 4 it is mentioned that the start and end of the eclipse are taken at 100 km altitude (which I suppose is also true in

the subsequent, similar figures). On the other hand, the sunrise indicator in Figure 4 apparently is at sea-level, which is not so relevant for the ionosphere. It would make more sense to consider what is going on at the altitude of the reflection of the signals (an eclipse can progress very different and various altitudes, see e.g. doi:10.1029/2020JA028088).

**Dear Anonymous Referee #1**. Thank you very much for this comment. Indeed, Figure 3 has been retrieved from https://eclipse.gsfc.nasa.gov/SEplot/SEplot2001/SE2019Jan06P.GIF where the solar eclipse moments are indicated at ground level. Regarding Figures 4–10, now, the captions under these figures include the following:
The vertical solid lines indicate, hereafter, the beginning, maximum phase, and the end of the solar eclipse at 100 km altitude, whereas dashed lines indicate sunrise at ground level because the sunrise at 100 km altitude occurred beyond the UT axis range, i.e., before 23:00 UT.

In this regard, it should also be considered that the various oblique links used are of different lengths, and operate with different frequencies (as described in Table 2). Thus, some oblique reflections might come from different altitudes than others. This could be a possible explanation why some links show less clear effects than others. It would be useful to have reflection heights listed in Table 2 as well (even if simply those expected from for instance IRI climatology; the reflection altitudes during the eclipse could be affected by the event itself).

**Dear Anonymous Referee #1**. Regarding the moments when sunrise occur at reflection altitudes, each of these altitudes is not known precisely; moreover, the eclipse acts to alter the reflection altitudes by tens of kilometers. In addition, the values of variations in the sunrise moments at different altitudes is insignificant as compared to the entire duration of the eclipse.
**Dear Anonymous Referee #1**. Thank you very much for suggesting the research topic of the altitudinal dependence of the solar eclipse effects. Now, we are going to invoke numerical modeling based on the results of (Verhulst and Stankov, 2020) and perform calculations of the ray paths to more precisely determine the reflection altitudes.

2) Further concerning the geometry of the eclipse, Table 3 lists the obscuration, magnitude and timing of the eclipse, again at 100 km altitude. However, these are presumably the obscuration in the visible light spectrum. For the ionosphere, the UV part of the spectrum is more relevant. Since the corona contributes a significant part of the solar EUV radiation, the values in Table 3 are not the most appropriate (see e.g. doi:10.1029/2017GL076771).

**Dear Anonymous Referee #1**. Thank you very much for this comment. Indeed, the corona contributes to the ionization of the atmosphere, and the value of this contribution could attain a factor of 0.2 the total rate of ionization, according to Mrak, S., Semeter, J.L., Drob, D., and Huba, J.D.: Direct EUV/X-ray modulation of the ionosphere during the August 2017 total solar eclipse. Geophys. Res. Let., 45, 3820–3828. https://doi.org/10.1029/2017GL076771, 2018.

We have estimated the contribution from the solar corona and have added the following text after Line 500 (marked in yellow):

The relations presented above could be specified by taking into account the ionization of atmospheric molecules with the ultraviolet emissions from the solar corona (Chernogor, 2013a), yielding

$$\frac{N_{\min}}{N_0} = \sqrt{\frac{1 - B_m + \xi}{1 + \xi}} \qquad\qquad (3)$$

where $\xi$ is a fraction giving the relative contribution of the solar corona to the ionization of the atmosphere. According to Mrak et al. (2018), the value of $\xi$ does not exceed 0.2, giving $\Delta N_{\max}/N_0 \approx -(10-16)\%$.

3) Finally, a comment concerning the introduction section. Between lines 70 and 105, a huge amount of references are given to earlier research into the ionospheric effects of solar eclipses. Obviously many such publications exist, but it is not clear what the precise relevance of each reference is to the current work. This paper is not intended to be a systematic literature review, so only those papers should be cited which are directly relevant for this work (. It should also be explained why they are relevant: e.g. in line 72 it is stated that certain papers should be noted, but why exactly are they relevant here?

**Dear Anonymous Referee #1**. Indeed, the presentation of the literature in the Introduction section looked bad. Generally, any author compiling an overview of, say 50, papers and indicating the main result of each study, tries to find an unresolved issue to be addressed in his or her paper. When writing this Introduction section, concerned with ionospheric effects of eclipses, the authors implied that, as Anonymous Referee #1 has rightly put it "there are no two identical reactions to two similar solar eclipses", and therefore, the Authors did not stated the results of any study under consideration, since the results of the current study inevitably provides new information, which supports the statement "there are no two identical reactions to two similar solar eclipses". The authors consider it appropriate to just mention the main works that are concerned with the ionospheric effects of solar eclipses. The effects are very varied. Regarding the solar eclipse of January 5–6, 2019, the authors were not aware of publications on this topic at the time of writing this paper.

To improve the presentation of the references, found between lines 70 and 105, the Authors have inserted the following introductory text:

The study of the ionospheric response to solar eclipses has advanced dramatically in the past 40 years. The important feature to note is that there are no two identical reactions to two similar solar eclipses. Therefore, we have only listed below the main works in the field.

**A few small technical comments:**

1) line 67: geospacer → geospace

**Dear Anonymous Referee #1**. We have corrected the misprint.

2) line 507: the maximum Doppler shift in Figure 8 appears to be over 0.5 Hz, rather than ~0.4 Hz

**Dear Anonymous Referee #1**. You are right. The maximum Doppler shift in Figure 8 is observed to be over 0.5 Hz. We corrected the text as follows (marked in yellow):

The Doppler shift exhibited a maximum shift, $\Delta f_{D\max}$, of ~0.4 Hz during the 23:40 UT to 00:10 UT period and a minimum shift, $\Delta f_{D\min}$, of 0.4 Hz during the 00:30 – 01:00 UT interval on the 5/6 January 2019 night.

The estimates of $(\Delta N/N_0)_{\max}$, as is shown further in the text, involve not just the maximum Doppler shift, $f_{D\max}$, but the shift in the maximum Doppler shift, $\Delta f_{D\max}$.

Thus, Figure 8 shows that the $f_{D\max}$ is observed to be over 0.5 Hz at 23:40 UT and 0.1 Hz at 00:10 UT, yielding ~0.4 Hz for $\Delta f_{D\max}$.

3) line 526: there is a deleted word "a" left here, which should read "the" instead.

**Dear Anonymous Referee #1**. We have corrected the misprint.

---

## Author Comment (AC2)

**RC2**: 'Comment on angeo-2022-15', Anonymous Referee #2, 30 Jun 2022
**Ionospheric effects of 5–6 January 2019 eclipse over the People's Republic of China: Results from oblique sounding**
Leonid Chernogor, Kostyantyn Garmash, Qiang Guo, Victor Rozumenko, and Yu Zheng
Reply to Anonymous Referee #2

Dear Anonymous Referee #2,

Thank you very much for your valuable comments that have helped the Authors greatly improve the draft of their paper.

Your comments are placed together with the Authors' answers, whereas the changes made in the text of the draft of the paper are marked in blue.

Authors.

**RC2**: 'Comment on angeo-2022-15', Anonymous Referee #2, 30 Jun 2022

This paper studied the ionospheric responses to the solar variations during the solar eclipse on January 6, 2019, by using the data of ionospheric oblique sounding from multiple propagation paths. The observation results are basically consistent with the theoretical estimates. There are no special new findings, and the article is too long. In addition, there are too many grammatical errors and redundant words. Therefore, it is suggested that the author should make necessary and sufficient modifications to the manuscript. The main comments are as follows:

1. Although this paper cited a lot of literature, it did not clearly and logically introduce the progress of these studies, but just listed them.

**Dear Anonymous Referee #2**. Indeed, the presentation of the literature in the Introduction section looked bad.

Generally, any author compiling an overview of, say 50, papers and indicating the main result of each study, tries to find an unresolved issue to be addressed in his or her paper. When writing this Introduction section, concerned with ionospheric effects of eclipses, the Authors implied that there are no two identical reactions to two similar solar eclipses, and therefore, the Authors did not stated the results of any study under consideration, since the results of the current study inevitably provides some new piece of information, which supports the statement mentioned above. Therefore, the authors consider it appropriate to just mention the main works that are concerned with the ionospheric effects of solar eclipses. The ionospheric effects are very varied.
          To improve the presentation of the literature, the Authors
(1) have combined some paragraphs and have added the topic sentences, marked in blue, "Some eclipses attracted particular attention" (Line 83), "More recently, increasingly sophisticated models have been developed" (Line 99);
(2) have inserted the following introductory text (Line 70):
The study of the ionospheric response to solar eclipses has advanced dramatically in the past 40 years. One should acknowledge that the manifestation of the ionospheric effects is dependent on many factors, including the measurement techniques. Thus, the involvement of techniques other than the conventional techniques (ionosonde, incoherent scatter radar, satellite radio beacon receivers, etc.) would be appropriate, including the technique used in this study. The important feature of the ionospheric response to note is that there are no two identical reactions to two similar solar eclipses. Therefore, the authors have restricted their review only to a listing of the main works in the field.

1. It is not necessary to make a very detailed description of the space environment on the day of the eclipse and the reference day. The authors just need to briefly describe that the geomagnetic activity is at a relatively low level before and after the solar eclipse. Therefore, the ionospheric changes can be attributed to the impact of the solar eclipse.

**Dear Anonymous Referee #2**. The suggestion to make just a brief description of the space environment somewhat differs from the geoscience community experience concentrated in the National Science Foundation Program on Coupling, Energetics and Dynamics of Atmospheric Regions (CEDAR). The last document "CEDAR: The New Dimension" states that the near-Earth environment is a "system that exhibits complexity – characterized by having multiple drivers, by featuring adaptive feedback and memory, by its nonlinear response and instabilities, and by exhibiting sensitivity to initial conditions," and further, "Aspects of this complexity include the importance of initial conditions, precondition, and memory; instability; nonlinearity; feedback; and emergent behavior."

Thus, a priori decision to analyze the geomagnetic activity before and after the solar eclipse alone should not be considered sufficient.

The authors have provided the briefest analysis of the state of space weather, and it is expedient. In addition, it is customary for other authors to do so.

If one placed the limitation "briefly describe that the geomagnetic activity is at a relatively low level", which the reviewer suggests, then the analysis would be unconvincing and unfounded.

2. It is not necessary to describe the results of each oblique probe in detail, just briefly describe the changes during the eclipse and summarize the common characteristics of these observations.

**Dear Anonymous Referee #2**. Regarding the results of each oblique probe, each oblique probe is a HF communication link which operates at its own frequency, have its own great-circle range of 910 km to 1,875 km (Table 2) and orientation shown in Figure 2. Each radio-wave propagation path can provide information on processes acting at propagation path midpoints (Table 2) at the reflection level. On the other hand, processes caused in the ionosphere by solar eclipses are altitude-dependent (Verhulst, T.G.W. and Stankov, S.M.: Height dependency of solar eclipse effects: the ionospheric perspective. J. Geophys. Res.: Space Phys., 125, e2020JA028088, https://doi.org/10.1029/2020JA028088 , 2020.). Therefore, for instance, Anonymous Referee #1 advises to employ more advanced techniques to analyze each propagation path. Moreover, the larger number of the propagation paths, the wider geographic and more extended altitude coverage of the event.

In the absence of illustrations and descriptions of the effects along different propagation paths, this work would provide unproven results. This is not acceptable in the scientific literature. Note that other reviewers consider that the usage of a large number of propagation paths provides a considerable advantage, in terms of encompassing both significant geographic regions and a wide range of altitudes.

3. There are too many paragraphs in this article, which need to be allocated more reasonably.

**Dear Anonymous Referee #2**. We have combined some paragraphs and marked them in red letters. Where needed, topic sentences have been added, marked in blue. For example, "Some eclipses attracted particular attention" (Line 83), "More recently, increasingly sophisticated models have been developed" (Line 99), or "The results may be summarized as follows:" (Line 562)

4. There are too many grammatical errors and redundant words. Please refer to the attached PDF file for specific English grammar comments.

**Dear Anonymous Referee #2**. We are sorry for making so many grammatical errors. Thank you very much for your great efforts to improve our English. The corrections we have made in the text are marked in blue.